# Lactoferrin Inhibits the Development of T2D-Induced Colon Tumors by Regulating the NT5DC3/PI3K/AKT/mTOR Signaling Pathway

**DOI:** 10.3390/foods11243956

**Published:** 2022-12-07

**Authors:** Huiying Li, Qianqian Yao, Chaonan Li, Linlin Fan, Haoming Wu, Nan Zheng, Jiaqi Wang

**Affiliations:** 1Beijing Key Laboratory of Food Processing and Safety in Forestry, College of Biological Sciences and Technology, Beijing Forestry University, Beijing 100083, China; 2Key Laboratory of Quality & Safety Control for Milk and Dairy Products of Ministry of Agriculture and Rural Affairs, Institute of Animal Sciences, Chinese Academy of Agricultural Sciences, Beijing 100193, China

**Keywords:** type 2 diabetes (T2D), colon cancer, lactoferrin (LF), 5′-nucleotidase domain-containing 3 (NT5DC3), phosphorylation

## Abstract

Although increasing evidence shows the association between type 2 diabetes (T2D) and colorectal cancer, the related mechanism remains unclear. This study examined the suppressive effect of lactoferrin (LF) on the development of T2D-induced colon cancer. First, a co-cultured cell model consisting of NCM460 and HT29 cells was constructed to mimic the progression of T2D into colon cancer. The migration ability of NCM460 cells increased significantly (*p* < 0.05) after cultivation in HT29 cell medium (high glucose), while LF suppressed the progression of T2D to colon cancer by regulating the 5′-nucleotidase domain-containing 3 (NT5DC3) protein and the PI3K/AKT/mTOR signaling pathway in diabetic BALB/c mice and in cell models. A mutation assay of the phosphorylation site in the NT5DC3 protein and a surface plasmon resonance (SPR) protein binding test were performed to further ascertain a mechanistic link between LF and the NT5DC3 protein. The results indicated that LF specifically bound to the NT5DC3 protein to activate its phosphorylation at the Thr6 and Ser11 sites. Next, metabolic-specific staining and localization experiments further confirmed that LF acted as a phosphate donor for NT5DC3 protein phosphorylation by regulating the downstream metabolic pathway in T2D-induced colon tumors, which was specifically accomplished by controlling Thr6/Ser11 phosphorylation in NT5DC3 and its downstream effectors. These data on LF and NT5DC3 protein may suggest a new therapeutic strategy for cancer prevention, especially in T2D patients susceptible to colon cancer.

## 1. Introduction

As a chronic metabolic disease, diabetes is characterized by hyperglycemia due to defective insulin secretion, impaired biological insulin action, or both [1] and usually results from environmental factors and genetic interactions. Approximately 425 million people worldwide suffered from diabetes in 2017, which is estimated to increase to 629 million by 2045 [2]. The long-term dietary intake of high glucose levels induces hyperglycemia, which is crucial to the pathogenesis of type 2 diabetes (T2D). Four trials that lasted more than half a year confirmed that a low dietary carbohydrate consumption was significantly associated with a reduction in the HbA1c concentration (an indicator for diabetes), indicating that dietary interventions and blood glucose metabolism improvement are significant for T2D onset [3,4,5,6].

Consistently high blood glucose levels cause damage to organs, such as liver, heart, and intestine, and recent studies have shown a strong link between T2D and colon cancer. Wu et al. (2013) reported that the risk of colorectal cancer was 1.22 times higher in diabetics than non-diabetics, which was consistent with studies conducted in various regions [7]. Additional research showed an increased risk of 30–40% for diabetic patients to develop colorectal cancer compared to non-diabetic ones [8,9]. The proliferation of colon cancer cells might be significantly promoted by elevated insulin and insulin-like growth factor-1 (IGF-1) levels, then increasing the vitality of transformed cells and ultimately inducing colorectal cancer [10]. Another possible mechanism involves the tumor-promoting role of leptin during hyperglycemia. In adipose tissue, excessive leptin levels activate the PI3K–MAPK signaling pathways, inhibiting apoptosis while enhancing angiogenesis and mitogenesis in colon tumor cells [11]. Therefore, controlling hyperglycemia may effectively inhibit the transformation of T2D into colorectal cancer.

Lactoferrin (LF) is an iron-binding glycoprotein with a variety of health-promoting activities, including anti-tumor and anti-diabetic properties [12]. Jolanta certified the effect of LF on hyperglycemic volunteers and found that LF decreased visceral fat tissue, as well as triglyceride and cholesterol concentrations in plasma, mainly through regulating IGF activity [13]. Research subsequently verified the hypoglycemic role of LF by enhancing insulin sensitization and immunomodulation in diabetic volunteers [14]. Moreover, LF could regulate LPS-mediated inflammatory cascades in a hypoglycemic environment [15]. The data indicated that LF supplementation ensured a more effective glycemic control than traditional anti-diabetic treatments alone. Therefore, it is speculated that LF may inhibit the transformation of T2D into colorectal cancer in a high glucose environment. To verify this hypothesis, the present study firstly investigated the ability of LF to inhibit the progression of T2D to colon cancer upon expoure to high glucose concentrations, both in vitro and in vivo, then confirmed the inhibitory role of the NT5DC3 protein during the progression of the two diseases. Later, the particular phosphorylation sites of the NT5DC3 protein regulated by LF were screened by High-Performance Liquid Chromatography (HPLC) and validated by mutation experiments, further confirming that LF suppressed the malignant development from T2D to colon cancer by activating NT5DC3 phosphorylation modifications.

## 2. Materials and Methods

### 2.1. Chemicals

Bovine milk LF (purified > 85%, Cat: L9507) was purchased from Sigma-Aldrich (St. Louis, MO, USA), while the NT5DC3 overexpression lysates (Cat: NBP2-07014) were bought from Novus Biologicals (San Diego, CA, USA). The antibodies against PI3K (Cat: K009329M), AKT (Cat: K109232P), mTOR (Cat: K002210P), p-PI3K (Cat: K006379P), p-AKT (Cat: K000186M), and p-mTOR (Cat: K006205P) were all purchased from Solarbio Life Science (Beijing, China). The antibodies against IL-1β (Cat: ab254360) and TNF-α (Cat: ab183218) were bought from Abcam (Shanghai, China).

### 2.2. NCM460 Cell Culturing in HT29 Cell Growth Medium

HT29 cells were cultured in DMEM medium containing 10% FBS and 1% penicillin/streptomycin in the lower Transwell^®^ chambers. Furthermore, the NCM460 cells were seeded into the upper Transwell^®^ chambers and cultivated in HT29 cell growth medium. All cells were cultivated at 37 °C in a 5% CO_2_ incubator.

### 2.3. Cell Viability and Invasion Assays

The NCM460 cells were seeded into the upper Transwell^®^ chambers and cultivated for 48 h in HT29 cell medium containing different glucose levels (0 g/L, 1 g/L, 2 g/L, 3 g/L, 4 g/L, 5 g/L, 6 g/L, 7 g/L, and 8 g/L). Then, the viability of the NCM460 cells was determined using CCK-8 to identify the appropriate glucose level. 

The HT29 cells were cultured in the lower Transwell^®^ chambers, while the NCM460 cells were seeded into the upper Transwell^®^ chambers, with 5 × 10^3^ cells in 200 μL of DMEM containing normal or high glucose concentrations. When the cells adhered to the wall, LF was added to the upper chamber, and the cells were cultured for 12 h. The migrated cells were observed using a microscope (Olympus, Tokyo, Japan) and counted. The above two experiments were repeated three times (*n* = 3).

### 2.4. Animal Models

The animal experiment comprised two parts. Part 1: 15 BALB/c mice were randomly divided into three groups, i.e., a control group (administered PBS), an LF group (administered 250 mg/kg b.w. of LF), and an NT5DC3 protein group (treated with 50 mg/kg b.w. of NT5DC3 injected via the tail vein), *n* = 5. Part 2: 15 BALB/c mice were fed a high-fat diet for 30 d, while their blood glucose was monitored to ensure the successful construction of the diabetic model. Then, the diabetic mice were assigned to three groups: a control group (administered PBS), an LF group (administered 250 mg/kg b.w. of LF), and an NT5DC3 protein group (treated with 50 mg/kg b.w. of NT5DC3 injected via the tail vein), *n* = 5. 

To construct the tumor model, HT29 cells were cultivated in 150 μL of Matrigel medium (BD) and injected into the backs of the mice. When the tumors reached 100–120 mm^3^, the mice were treated with PBS, LF or NT5DC3. At the end of the experiment, the mice were sacrificed, and the tumor volumes were measured.

The animal experiments were approved by the Ethics Committee of the Chinese Academy of Agricultural Sciences (Beijing, China; permission number: IAS2020-90).

### 2.5. Western Blot Detection

NCM460 cells were lysed with RIPA buffer, and the total protein concentration was measured using a BCA kit (Beyotime, Nantong, China). The antibodies included anti-human NT5DC3 (PA5-70919, Invitrogen, Waltham, MA, USA) and anti-human β-actin (PAB0865, Enzo Life Sciences, New York, NY, USA). For western blotting, the primary antibodies were diluted 1:1000, while the secondary antibodies were used at a 1:3000 dilution. The signals were captured and analyzed using Clinx ChemiCapture software (Clinx, Shanghai, China). This experiment was repeated three times (*n* = 3).

### 2.6. Co-Immunoprecipitation Mass Spectrometry (CoIP–MS) Analysis

The CoIP-MS analysis was conducted in accordance with existing references with appropriate adjustments. HT29 cells exposed to different treatments (normal glucose, high glucose, normal glucose + LF, high glucose + LF, high glucose–normal glucose, and high glucose–normal glucose + LF) were lysed with Triton X-100 lysis buffer supplemented with a protease inhibitor cocktail. The lysates were centrifuged at 5000 g for 20 min at 4 °C, after which the supernatants were centrifuged again at 20,000 g for 10 min at 4 °C. The total protein concentrations of the samples were measured using the BCA assay. The NT5DC3 antibody was added, after which NT5DC3 and its interacting proteins were purified with antibody-conjugated beads. Then, the immunoprecipitates were washed with ice lysis buffer and incubated with 2× sample buffer to obtain the enriched CoIP products (www.biosignaling.biomedcentral.com (accessed on 01/08/2022).), which were analyzed via mass spectrometry (2 h). The appropriate NT5DC3 phosphorylation sites regulated by LF were further screened via data analysis (Appendix A). This experiment was repeated three times (*n* = 3).

### 2.7. Phosphorylation Site Mutation Validation

A phosphorylation site mutation assay was conducted to validate the function of the possible NT5DC3 phosphorylation regions of interest (ROI), namely, Thr6 and Ser11. Three types of DNA sequences (details can be found in the Supplementary mutant sequence design of the NT5DC3 protein in Appendix A phosphorylation sites mutation) were synthesized (RuiBiotech, Beijing, China), i.e., wild type, with the consensus coding sequence (CCDS) of NT5DC3; Thr6-mut, in which the 68th codon for Thr (ACA) was changed into a codon for Pro (CCC); Ser11-mut, in which the 206th codon for Ser (AGC) was changed into a codon for Pro (CCC). The expression plasmids encoding NT5DC3 and its mutations were further generated by cloning using the pcDNA3.1(+) vector, subsequently transformed into DH5α competent cells. The constructed pcDNA3.1(+) plasmids were extracted from the positive clones and confirmed via sequencing. Next, a mutation experiment was performed, and the plasmids were transfected into HT29 cells to validate the phosphorylation sites and further elucidate the role of NT5DC3 phosphorylation during the progression of T2D to colon cancer. 

### 2.8. MS and Matrix-Assisted Laser Desorption Ionization (MALDI) Imaging

Liver, cancerous, and paracancerous mouse tissue were used for molecule identification. Here, 10-μm-thick tissue sections were prepared using a freezing microtome (Leica, Wetzlar, Germany) and mounted on ITO-coated glass slides for MALDI–IMS. The matrix 2,5-dihydroxybenzoic acid (DHB) was sublimated using an iMLayer vacuum deposition system (Shimadzu, Kyoto, Japan). Gaseous DHB was deposited onto the specimen surface at a thickness of 1.5 μm. MALDI–MS and imaging MS were performed using a SHIMADZU iMScope TRIO. Optical images of sections were acquired using an iMScope before sample preparation. The tissue sections were scanned at a resolution ratio of 50 μm × 50 μm in positive ion mode, a sample voltage of 3.5 kV, and a detector voltage of 1.75 kV. The MS data were acquired in a mass range of *m*/*z* 400–900 and 50% laser power (SHIMADZU notation). The variation in the different sections was compared using ImagingMass Solution according to the optical and MSI data images. For MS/MS analysis, the sections were scanned at a resolution ratio of 100 μm × 100 μm in positive ion mode, a sample voltage of 3.5 kV, and a detector voltage of 1.80 kV. The MS data were acquired in a mass range of *m*/*z* 100–850. The compound was further analyzed at a parent ion of 798.54. This experiment was repeated three times (*n* = 3).

### 2.9. Pathological Detection by HE Staining

Liver, tumor, and paracancerous tissues were isolated and fixed in 4% paraformaldehyde for 24 h. The tissue samples were embedded in paraffin and sliced using a slicing machine (Leica). Hematoxylin and eosin were used to stain the slices, which were observed using a light microscope (Olympus). The experiments were repeated three times

### 2.10. Binding Detection between LF and the NT5DC3 Protein

To determine the binding affinity between LF and the NT5DC3 protein, a surface plasmon resonance (SPR) assay was performed using a BIAcore T200 instrument (GE Healthcare, Chicago, IL, USA) equipped with BIAcore sensor chips (GE Healthcare), while the data were analyzed with a matching BIAcore control software. The instrument flow path was washed, and the chips were activated according to existing references [16]. A 50 μg·mL^−1^ (0.91 μM) sample of purified NT5DC3 protein lysate (purity above 98%, Novus Biologicals) was amino-immobilized onto a CM5 chip in 10 mM sodium acetate buffer to achieve a density of 10 kRU and subsequently blocked using ethanolamine hydrochloride (Thermo Fisher Scientific, Waltham, MA, USA). A series of LF protein solutions at different concentrations (1.2 μM, 0.3 μM, 0.075 μM, 0.019 μM, and 0.0047 μM) were separately injected into the flow path at a flow rate of 5 μL·min^−1^, a contact time of 120 s, and a dissociation time of 150 s. A regeneration step was not conducted. Finally, the association rate (ka) and dissociation rate (kd) were fitted. This experiment was repeated three times (*n* = 3).

## 3. Results

### 3.1. LF Inhibited the Proliferation and Migration of NCM460 Cells Cultured in Colon Cancer Cell Medium

We constructed an NCM460 and HT29 cell co-culturing model, in which HT29 cell growth culture medium was further utilized to culture NCM460 cells (Figure 1A). As shown in Figure 1B, NCM460 cell viability exceeded 90% at a glucose concentration from 2 g/L to 5 g/L and was suppressed at glucose levels beyond this range. Therefore, 2 g/L and 5 g/L were chosen as the normal and high glucose levels, respectively. Moreover, the migration ability of NCM460 was remarkably inhibited by LF compared to that of the control group (*p* < 0.05) in both normal and high glucose conditions (Figure 1C,D).

### 3.2. LF Suppressed Tumor Growth in T2D Mice

The HT29 cells were transplanted into diabetic and non-diabetic mice exposed to T2D treatment and then treated with LF and NT5DC3. The results showed that both LF and NT5DC3 significantly suppressed tumor growth in diabetic mice compared with non-diabetic mice (*p* < 0.05) (Figure 1E,F).

### 3.3. LF Regulated NT5DC3 Expression in NCM460 Cells (with HT29 Cell Culturing Medium) in High Glucose Conditions

To investigate the mechanism of LF suppressing tumor growth in a high-glucose environment, we measured the expression of NT5DC3/PI3K/AKT/mTOR proteins in NCM460 cells co-cultured with HT29 cells in normal or high glucose conditions. The expression of NT5DC3 declined significantly during hyperglycemia, while that of other proteins, such as p-PI3K, p-AKT, p-mTOR, IL-1β, and TNF-α, was substantially higher. However, this tendency was reversed when the cells were treated with LF. The p-PI3K/PI3K, p-AKT/AKT, and p-mTOR/mTOR ratios were all lower after LF supplementation in a high-glucose environment (Figure 2A,B). The findings suggest that LF may regulate tumor progression during hyperglycemia by regulating the NT5DC3/PI3K/AKT/mTOR signaling pathway.

### 3.4. LF Activates NT5DC3 Phosphorylation

All the above results suggested that NT5DC3 could mediate the development of colon cancer in diabetic patients under regulation of LF. To further ascertain the role of NT5DC3, a phosphorylation site mutation assay was conducted. Threonine and serine residues represent primary phosphorylation sites. Point mutation to more stable amino acids can be applied to examine the phosphorylation site-specific function. The phosphorylation sites in NT5DC3 were identified as Thr6 and Ser11 via Co-IP and MS co-analyses (Figure 3A,B, Appendix A), while further assessing the specific phosphorylation sites in the NT5DC3 protein.

TheNT5DC3 protein presents multiple phosphorylation sites, which are activated differently in various types of cell lines (http://qphos.cancerbio.info/show.php? (accessed on 1 August 2022). type = advanced), while its expression level can denote the phosphorylation conditions. In this study, we constructed NT5DC3 site-mutant fragments at different phosphorylated sites (Appendix A: Mutant sequence design of the NT5DC3 protein, in Appendix A phosphorylation sites mutation), which were introduced into HT29 cells in different glucose conditions. Then, cell migration and invasion assays, as well as Western blotting detection of the NT5DC3 protein, were carried out to evaluate the effect of LF on HT29 cells with the Ser11/Thr6 mutants. The results demonstrated that LF could not inhibit the migration and invasion of mutant HT29 cells compared to the control group without any treatment. In the mutant groups, no significant differences were evident in the number of invaded cells and recovery rates between the normal and normal + LF groups or the high-glucose and high-glucose + LF groups (Figure 3C–F). Moreover, LF failed to upregulate the NT5DC3 protein level in both normal and high glucose conditions (Figure 3G), confirming that LF inhibited HT29 cells migration and invasion, mainly via the regulation of Thr6/Ser11 phosphorylation in NT5DC3. The findings of this study demonstrated for the first time the regulatory effect of LF on NT5DC3 expression from three epigenetic perspectives.

### 3.5. Special Metabolites Screened via Atmospheric Pressure-MALDI–MS (AP-MALDI–MS)

We then conducted an AP-MALDI–MS assay to screen special metabolites necessary for NT5DC3 to exert its function. Liver, tumor, and paracancerous tissues were identified and selected for imaging according to information related to significantly changed *m*/*z* values via AP-MALDI–MS, using MS/MS data and corresponding literature (Figure 4A). Ion images were reconstructed to reveal their distribution on the tissue sections. The *m*/*z* 798.54 ([PC(34:2)+K]^+^ (Width: 1, Energy: 32)) in the liver, tumor, and paracancerous tissue showed a distinctly lower expression in the LF and NT treated groups than in the untreated groups (Figure 4B), which might be related to the exposure of LF and its metabolic mechanism. An ROI analysis of the tissues from the untreated and LF/NT exposed groups was conducted, showing significant differences between the untreated and LF-/NT-exposed groups (Figure 4C). Additionally, the AP-MALDI–MS spectra obtained from a tumor tissue section in positive ionization mode, shown in Figure 4D, indicated the possible intermediate metabolism of [PC(34:2)+K]^+^. These results indicated that an alternative mechanism may be involved, besides the regulatory effect exerted by LF or its related metabolites, such as [PC(34:2)+K]^+^ (as a phosphate group supplier), on NT5DC3 protein phosphorylation.

### 3.6. LF Binds to the NT5DC3 Protein

In order to further identify the mechanistic link between LF and NT5DC3, SPR assays were performed to measure the binding affinity between LF and the potential target NT5DC3 protein. BIAcore sensor chips were coated with the purified NT5DC3 protein before exposure to LF at different concentrations. The NT5DC3 group results revealed an association constant (ka, 1/Ms) of 9.70 × 10^4^, a dissociation constant (kd, 1/s) of 1.03 × 10^−5^, and an equilibrium dissociation constant (KD) of 1.06 × 10^−8^ M (Figure 5). Therefore, these findings suggest that LF may inhibit the progression of T2D to colon cancer by directly binding to the NT5DC3 protein.

## 4. Discussion

Increasing evidence shows an association between T2D and various types of cancer, including breast cancer, pancreas cancer, liver cancer, and colon cancer [17,18]. Tsilidis et al. (2015) summarized the association between T2D and cancers and revealed that the risk of developing cancer was higher in diabetic patients than in non-diabetic individuals [19]. Mοreover, the incidence of colorectal cancer increases by 20% in the presence of hyperglycemia [20]. Although the mechanism behind the tendency of T2D patients to develop colon cancer remains unclear, several possible hypotheses have been proposed. (a) Changes in metabolites in T2D patients, such as in insulin/IGF-1 and leptin/adiponectin secretion, may favor tumor development. Persistently low IGF-1 concentrations in T2D patients induce glucose metabolism disorders and cytokine destruction, affecting tumor metabolism pathways [21]. (b) Severe inflammatory responses in T2D patients facilitate cancer development. (c) Hyperglycemia induces epigenetic changes, such as DNA methylation, and leads to abnormal gene expression, further influencing the proliferation of tumor cells [22]. Although the mechanism underlying diabetes-induced colon cancer has elicited considerable debate, hyperglycemic control might effectively inhibit the transformation of T2D to colorectal cancer, which is crucial for suppressing T2D-susceptible gastrointestinal tumors.

LF is present in the colostrum and mature milk of most mammals and is considered an important host defense molecule. LF levels significantly increase when the body is infected by external pathogens. In addition, it fulfills a variety of other biological functions, presenting antioxidant, anti-inflammatory, anti-cancer, and immune regulatory properties [23]. The anti-tumor ability of LF has been widely confirmed in various in vivo and in vitro tumor models [24,25]. Recent research revealed the ability of LF to alleviate T2D, improving hepatic insulin resistance and pancreatic dysfunction by regulating the PI3K/AKT signaling pathway in diabetic mice [26]. A clinical test involving 60 young T2D patients demonstrated the anti-diabetic efficacy of LF. The 60 subjects were treated with oral LF capsules (250 mg/day, p.o.) for three months. The results indicated that LF exerted an insulin-sensitizing and anti-inflammatory effect by regulating the TLR4/NFκB/SIRT-1 pathway [14]. To explore the underlying mechanism of LF in suppressing diabetically induced colon cancer, this study constructed a co-culture model consisting of NCM460 and HT29 cells. After treatment with LF, the migration and invasion abilities of NCM460 cells in a high-glucose medium were significantly inhibited. In addition, LF significantly restrained tumor growth in diabetic mice compared with non-diabetic mice. These results suggested that LF could inhibit malignant lesions in the intestine in a high-glucose environment.

Furthermore, NT5DC3, a member of the NT5DC family, is an evolutionarily conserved 5′-nucleotide enzyme that catalyzes intracellular nucleotide hydrolysis and contains a haloacid dehalogenase motif [27]. Currently, NT5DC2 is regarded as a biomarker for colorectal carcinoma since it regulates multiple cellular events to modulate tumor growth [28]. Since reports involving the role of NT5DC3 in colorectal cancer are rare, it is necessary to examine it as a key risk factor. The present study examined the role of NT5DC3 in colorectal tumor growth. The results indicated that NT5DC3 efficiently suppressed the development of T2D to colorectal tumors in BALB/c mice, while NT5DC3 protein expression declined during hyperglycemia, and LF inhibited this malignant progression by regulating the levels of NT5DC3 and the PI3K/AKT/mTOR signaling pathways. Since the data showed that NT5DC3 was involved in the process of T2D development into colorectal tumors, and phosphorylation is considered an important protein modification to ensure normal physiological function [29,30], to investigate the mechanistic link between LF and NT5DC3 protein, the phosphorylation sites in NT5DC3 protein were screened by HPLC assay and confirmed by mutation assay, and the binding affinity between LF and NT5DC3 protein was measured by SPR. The results showed that LF displayed a specific binding reaction with NT5DC3 protein, by activating NT5DC3 protein phosphorylation at the Thr6 and Ser11 sites. After mutating Ser11/Thr6, the NT5DC3 protein acquired the ability to inhibit the migration and invasion of HT29 cells. Furthermore, metabolic-specific staining and localization experiments further confirmed that LF might be a phosphate donor for NT5DC3 protein phosphorylation by regulating downstream metabolic intermediates [PC(34:2)+K]^+^.

In conclusion, the present study verified the inhibitory effect of LF in the development of T2D colon tumors by regulating Thr6/Ser11 phosphorylation in the NT5DC3 protein and its downstream factors. LF regulation of this pathway may represent a new method for cancer prevention, especially in T2D patients.

## Figures and Tables

**Figure 1 foods-11-03956-f001:**
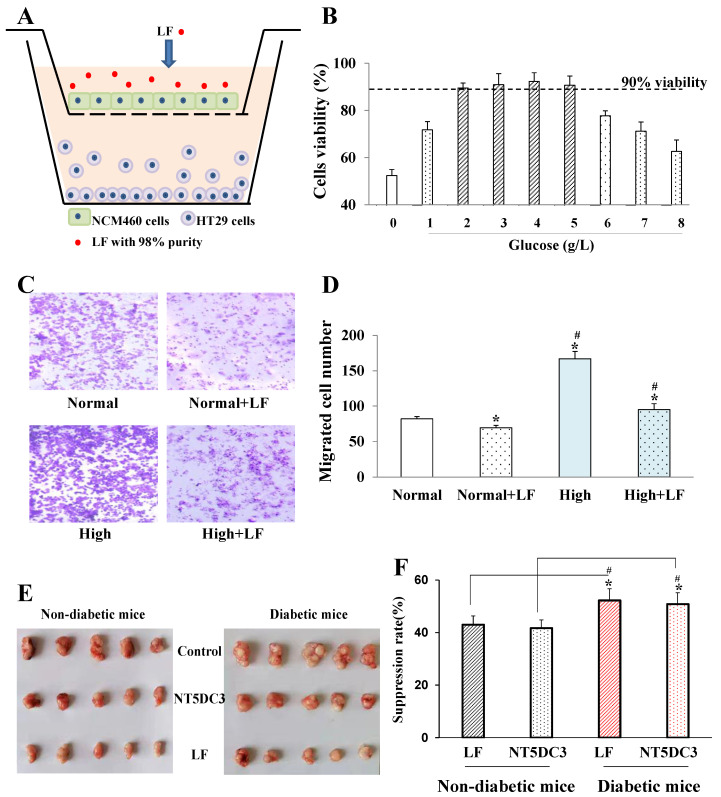
The anti-tumor effect of lactoferrin (LF) and NT5DC3 in high glucose conditions. (**A**) Co-culturing model consisting of NCM460 cells and HT29 cells. HT29 cell growth culture medium was further utilized to cultivate the NCM460 cells. (**B**) Glucose concentration selection using the NCM460 cells and a cell viability assay. Here, 2–5 g/L of glucose was regarded as the appropriate concentration for subsequent experiments. (**C**) Invasion ability assay of NCM460 cells in the presence of different glucose concentrations, using Transwell (200×). (**D**) Number of migrated cells in (**C**). (**E**) Tumors in the different non-diabetic and diabetic mouse groups. (**F**) Suppression rate in each group in (**E**). The data are presented as mean ± SD, * *p* < 0.05 compared with the LF or Non-diabetic NT5DC3 group, ^#^
*p* < 0.05 compared with the normal group in (**D**), ^#^
*p* < 0.05 compared with the normal group in (**F**). (**A**–**D**) the experiments were repeated three times (*n* = 3); (**E**,**F**) the experiments were repeated five times (*n* = 5).

**Figure 2 foods-11-03956-f002:**
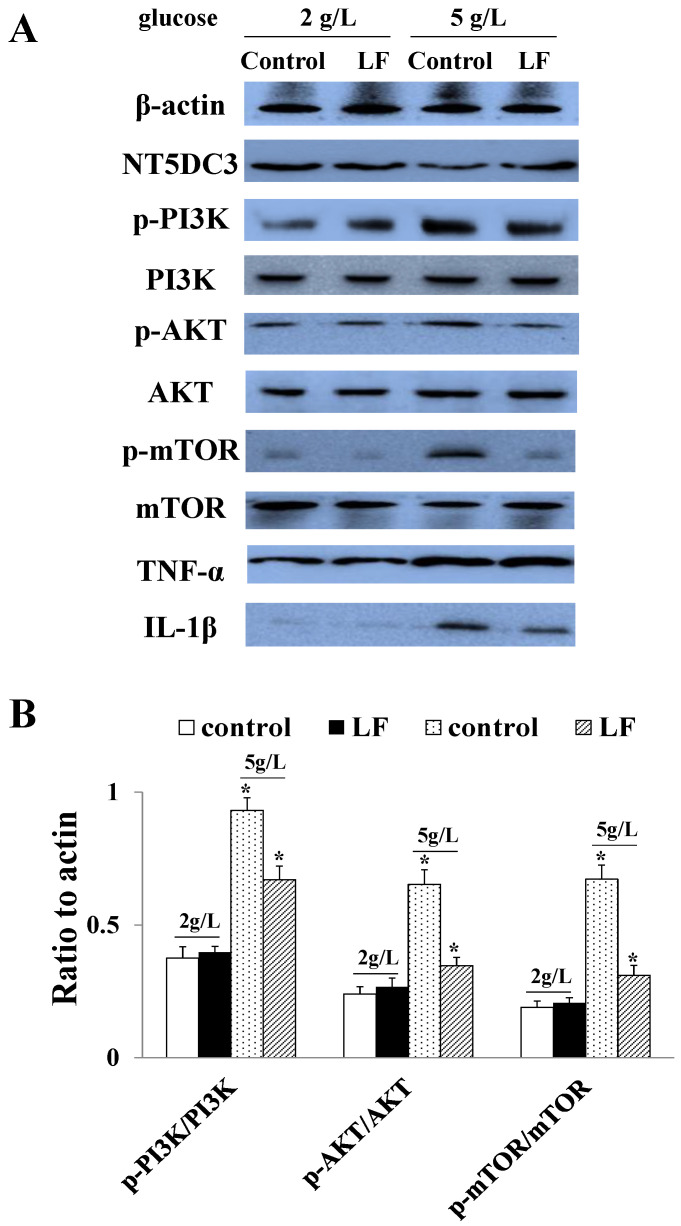
Expression levels of the proteins in NCM460 cells co-cultured with HT29 cells at different glucose concentrations. (**A**) Protein bands of NT5DC3, p-PI3K, PI3K, p-AKT, AKT, p-mTOR, mTOR, TNF-α, and IL-1β, determined via Western blotting. (**B**) Statistical analysis of the p-PI3K/PI3K, p-AKT/AKT, and p-mTOR/mTOR ratios. The data are presented as mean ± SD, * *p* < 0.05 compared with the control group. This experiment was repeated three times (*n* = 3).

**Figure 3 foods-11-03956-f003:**
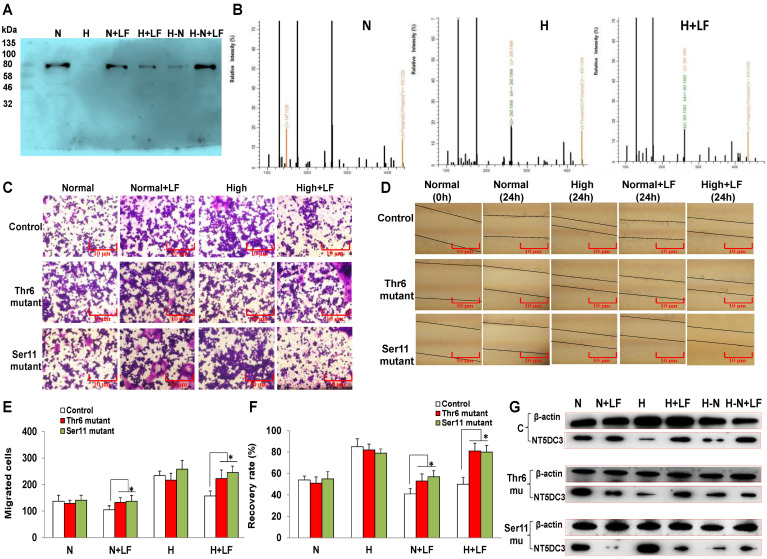
Phosphorylation site investigation in the NT5DC3 protein. (**A**) CoIP gel of the NT5DC3 protein, obtained via Western blotting. (**B**) Phosphorylation at Thr6 and Ser11, determined via secondary mass spectrometry. (**C**) Images of the invaded HT29 cells in the Transwell experiment p(200×). (**D**) Images of the migrated HT29 cells obtained via a scratch assay (200×). (**E**) Statistical analysis of the invaded HT29 cells. (**F**) Statistical analysis of the migrated HT29 cells. (**G**) NT5DC3 protein expression in the HT29 cells obtained via Western blotting. The data are presented as mean ± SD, * *p* < 0.05 compared with the control group. This experiment was repeated three times (*n* = 3).

**Figure 4 foods-11-03956-f004:**
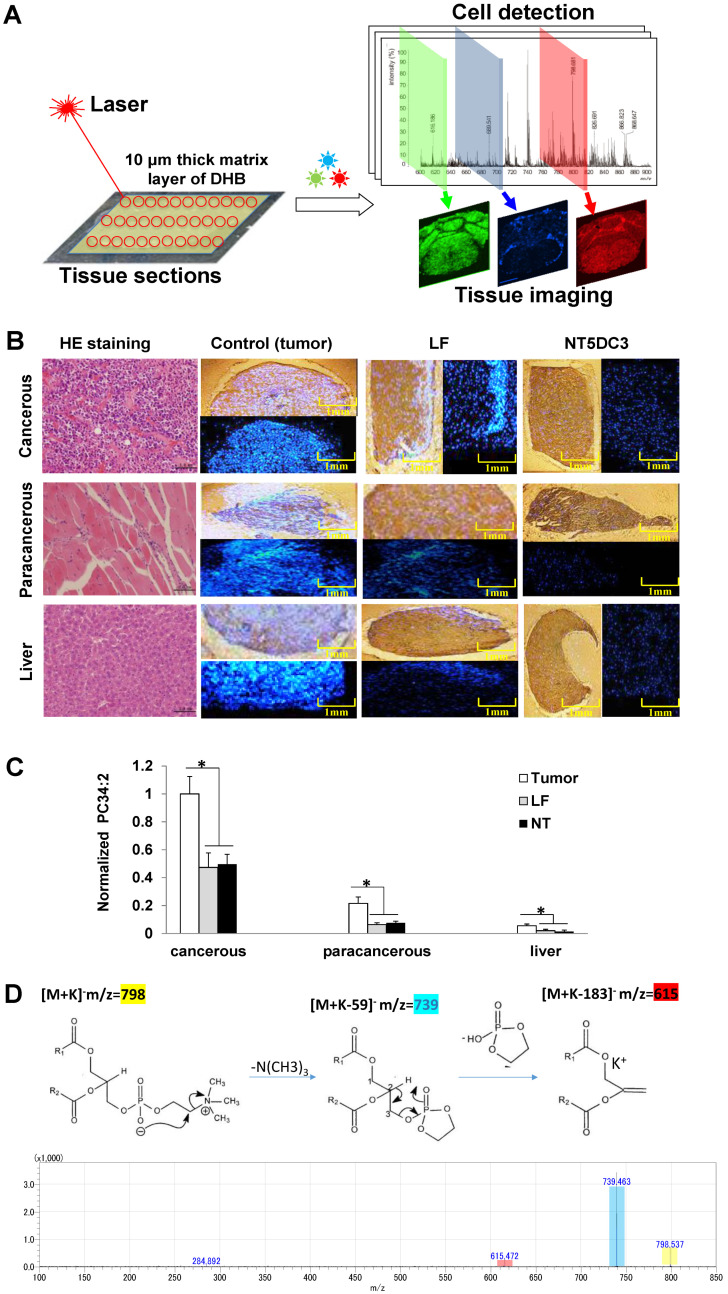
Special metabolite screening via AP-MALDI–MS. (**A**) Workflow AP-MALDI–MS imaging. (**B**) HE staining results (40×) and MSI of the cancerous, paracancerous, and liver tissues in positive ionization mode after LF/NT exposure and no treatment, *m*/*z*: 100–850. (**C**) ROI analysis of the three types of tissue in the untreated and LF-/NT-exposed groups via the normalization of [PC(34:2)+K]^+^. A *t*-test was conducted using Imaging MS solution Ver.1.30. (**D**) AP-MALDI–MS/MS spectra of *m*/*z* 798.54 acquired from the tumor tissue sections using the DAN matrix in positive mode. Yellow color label stands for the selected metabolite, blue color label stands for the primary decomposition product of the metabolite, red color label stands for the secondary decomposition product of the metabolite. The data are presented as mean ± SD, * *p* < 0.05 compared with the tumor group. This experiment was repeated three times (*n* = 3).

**Figure 5 foods-11-03956-f005:**
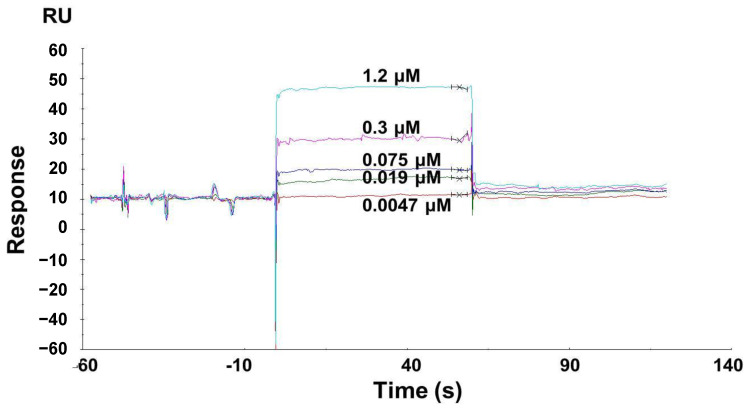
The affinity between LF and the NT5DC3 protein, determined via SPR. This experiment was repeated three times (*n* = 3).

## Data Availability

Not applicable.

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
