# Peer review of "Lactoferrin Inhibits the Development of T2D-Induced Colon Tumors by Regulating the NT5DC3/PI3K/AKT/mTOR Signaling Pathway"

_foods, 2022, doi:10.3390/foods11243956_

Round 1
Reviewer 1 Report
The manuscript by Qianqian Yao et al. “Lactoferrin inhibits the development of T2D to colon tumors 2 by regulating NT5DC3/PI3K/AKT/mTOR signaling pathway” examined the suppressive effect of lactoferrin on the development of type 2 diabetes (T2D)-induced colon cancer.
The authors demonstrated that lactoferrin specifically bound with the NT5DC3 protein to activate its phosphorylation at the Thr6 and Ser11 sites of NT5DC3. In addition, the metabolic-specific staining and localization experiments confirmed that lactoferrin provided a phosphate donor for NT5DC3 protein phosphorylation by regulating the downstream metabolic pathway in T2D-induced colon tumors. This was specifically accomplished by controlling the Thr6/Ser11 phosphorylation of NT5DC3 and its downstream factors. The authors conclude that this inhibitory effect of lactoferrin on development of T2D-induced colon tumors may represent a new method for cancer prevention, especially in patients with T2D.
Together, this is a very interesting manuscript that presents original data concerning the protective effects of lactoferrin on colon cancer induced by T2D.
Some minor aspects are raised in the manuscript that should be addressed by the authors.
S 1. Some figures need to have their quality improved, such as figures 1d, 3a, 4b, 4c and 4d. For example, in figure 4B, the resolution of the images is poor, the magnification bars are located over the images, and there are dark areas devoid of images.
2 2. In figure legend 3, please change “the normal group” by “the control cells”; and the authors do not indicate the meaning of the asterisk in the figure legend 4.
3 3. Some linguistic mistakes should be corrected and the whole manuscript should be revised by native English speaker.
Author Response
Some minor aspects are raised in the manuscript that should be addressed by the authors.
1 Some figures need to have their quality improved, such as figures 1d, 3a, 4b, 4c and 4d. For example, in figure 4B, the resolution of the images is poor, the magnification bars are located over the images, and there are dark areas devoid of images.
Answer:Thank you so much for your suggestions. We have checked the figures, and replaced them with the improved ones, which are demonstrated in the attachment.
2 In figure legend 3, please change “the normal group” by “the control cells”; and the authors do not indicate the meaning of the asterisk in the figure legend 4.
Answer:Yes, we have corrected “the normal group” to “the control cells” in Figure 3 legend, and we have added the explanation of the meaning of the asterisk in Figure 4 legend, as “The data are presented as mean ± SD, * p < 0.05 compared with the tumor group (n = 3)”.
3 Some linguistic mistakes should be corrected and the whole manuscript should be revised by native English speaker.
Answer:Yes, we have asked a native English speaker to revise the manuscript, to make our expression more clear and comprehensible.
To sum up, thank you very much for your meaningful suggestions and help!

Reviewer 2 Report
The manuscript entitled as “Lactoferrin inhibits the development of T2D to colon tumors by regulating NT5DC3/PI3K/AKT/mTOR signaling pathway” is very interesting. The results are clearly depicted, elucidated and explained. How every I have few suggestions as described below:
1. I think the densitometry represented in figure 2B does not seems to match with the representative images, particularly in 2 g/L of glucose.
2. How many times the experiment was conducted? Please provide the data.
3. Authors need not to present the individual protein, instead that can show the ratio of phosphorylated form/total protein.
4. In figure 3C, there is an image duplication. The control and ser11 normal are from same well, they are just of different regions. This is not acceptable in any circumstances. authors need to provide the raw data to justify (replicates).
Author Response
1 I think the densitometry represented in figure 2B does not seems to match with the representative images, particularly in 2 g/L of glucose.
Answer:Thank you so much for your suggestion! We have checked the scanning data of gel bands in 2 g/L-glucose group, and find somethings wrong here. We have corrected the figures in Figure 2, as shown in the attachment.
2 How many times the experiment was conducted? Please provide the data.
Answer:Yes, we have added the information of experiment repetitions in “Materials and Methods” and “Figure legends” parts, to make our expression more clear.
3 Authors need not to present the individual protein, instead that can show the ratio of phosphorylated form/total protein.
Answer:Yes, we have deleted Figure 2B here, and the ratios of phosphorylated protein/total protein were demonstrated in Figure 2C.
4 In figure 3C, there is an image duplication. The control and ser11 normal are from same well, they are just of different regions. This is not acceptable in any circumstances. authors need to provide the raw data to justify (replicates).
Answer:We must apologize for our carelessness and mistakes, we are so sorry. I have searched the raw data and carefully checked the images in Figure 3C, and corrected this mistakes by replacing the two figures (the control and the ser11 in normal group). The figures from raw data are shown in the attachment.
To sum up, thank you very much for your meaningful suggestions and help!

Round 2
Reviewer 2 Report
The current form of the manuscript looks fine to me after revision. I am glad that authors have addressed the questions which were raised and solved the problems. Therefore, I believed the present format of the manuscript can be accepted at this point.